# Business Models Amid Changes in Regulation and Environment: The Case of Finland–Russia

**Oskari Lähdeaho [1] and Olli-Pekka Hilmola [1,2,*]**

[1]   Industrial Engineering and Management, LUT Kouvola, LUT University, Prikaatintie 9, FIN-45100 Kouvola, Finland; oskari.lahdeaho@lut.fi

[2]   Estonian Maritime Academy, Tallinn University of Technology (Taltech), Kopli 101, 11712 Tallinn, Estonia

*   Correspondence: olli-pekka.hilmola@taltech.ee

**Abstract:** Changes in regulation are affecting the international business environment. In this study the impact of regulation changes and ways to benefit from those in Finland and Russia are examined. Logistics and manufacturing companies are studied using the case study approach including ten semi-structured interviews (Finland and Russia) and a survey (Southeast Finland), further supported by an additional survey for logistics sector companies (Southeast Finland). The changes in the business environment have created a fragmented market with a growing number of actors. Three business models (blockchain-based, platform-based and innovative subcontracting-based), capitalizing on the growing number of actors, were incepted in the interview phase and evaluated in the survey phase with companies. These models are integrable with the circular economy, a relevant practice according to the studied companies. Blockchain was perceived as a still immature technology. Further study revealed that the companies are not well prepared for environmental demands in logistics, and the overall volumes and business climate between the analyzed countries have not improved. Additionally, those companies do not actively pursue the possibilities of new technologies. The impact of regulatory changes in this region has not been examined closely with a case study approach. This study helps to explain the current trends in an established market.

**Keywords:** business models; regulation; logistics; supply chains; Finland; Russia

## 1. Introduction

The international railway connection between Finland and Russia (then Union of Soviet Socialist Republics) was formalized in 1948 when these countries signed a contract on interconnection via railways [1]. Since then, policymaking between these countries has progressively been facilitating international trade, and furthermore connecting eastern and western markets. More recent changes in legislation and regulation have been gone along the lines of European Union's (EU) railway legislation renewal. These so-called railway packages seek to open railway transports for competition and to enable fluent international operations [2]. However, some restrictive legislative changes have also emerged between Finland and Russia, driving the affected industries to adapt to the risen challenges. Thereupon, the changing business environment requires involved companies to revise their respective business models.

Recent turbulence in the political environment between the EU and Russia has led to sanctions by the EU and respectively counter sanctions by Russia. The sanctions affect the exports to and imports from Russia of military equipment, dual-use goods, energy-related equipment (mainly concerning oil exploration and production) and supporting services for the mentioned equipment and goods [3], whereas the counter sanctions are mainly imposed against the import of food products (such as

milk, dairy products and meat), but also various other industries and sectors of the economy [4], e.g., agricultural equipment and pharmaceutical goods [5].

Recently, the EU has introduced strategies and regulations to lower emissions produced by transport, which is globally responsible for a major share of all produced emissions, i.e., approximately a quarter of all carbon dioxide emissions [6]. The general goal for carbon dioxide emissions from transport is to reduce them by 2050 to 60% lower in comparison to the base year of 1990 [7]. Furthermore, the EU has committed to reducing other emissions that also occur from transport by 2020 and beyond in comparison to 2005 levels, namely sulfur (by 59%), nitrogen oxides (by 42%), ammonia (by 6%), volatile organic compounds (by 28%) and atmospheric particulate matter (diameter less than 2.5 micrometers; by 22%) are targeted [8]. In addition, the International Maritime Organization (IMO) has set the cap of sulfur content in fuels used by ships to 0.5% starting from January 2020 [9]. A more demanding sulfur regulation of 0.1% was implemented in the Baltic and North Sea already in 2015 [10]. Moreover, nitrogen oxide emissions are strictly controlled by the IMO in the Baltic and North Sea emission control area with a reduction target of 80% in comparison to 2016 by requiring all the new ships built after January 2021 to have either installed catalyst converters for emissions or use Liquefied Natural Gas (LNG) as their fuel [11]. IMO also prohibits the discharge of waste from maritime vessels into the sea and requires monitoring of the waste while on-board until proper disposal [12]. These type of regulations incentivizes companies to optimize their supply chains and transport mode selection correspondingly [13]. Moreover, while the presented regulations cover specific regions, the environmental policies can be expected to be implemented in other areas depending on the policies' effectiveness and the given area's economic and political situation [14]. Therefore, adapting to emission regulations is vital for companies operating internationally.

While the presented legislation changes tighten the business environment for the involved companies, the transformed environment could facilitate or even necessitate the deployment of new business models. Thus, this article seeks to provide insight into the following questions:

- RQ1: What effects do legislative changes have on the business environment between Finland and Russia?
- RQ2: What types of new business models are enabled by the changing environment?
- RQ2.1: How are innovation and new technologies enabling these business models?

The rest of the article follows a structure of a literature review on sustainability in supply chains and logistics as well as sustainable business model innovation in Section 2, succeeded by explaining the used methodology to study the impact of the changing business environment in Southeast Finland and its implications for manufacturing and transportation companies in Section 3. Thereafter, the results of the empirical study are presented in Section 4. Lastly, this article is concluded in Section 5, where the results are discussed and reflected upon the theoretical baseline, and possible directions for future research are drawn.

## 2. Sustainable Business Models in Logistics

While sustainability has been established as an essential topic in supply chain research [15], generally it remains inadequately regarded in practical supply chain operations [16]. A modest shift of the aspects creating value in logistics has been observed, from a rigid cost orientation to other factors, environmental sustainability being one [17,18]. In maritime traffic, environmental sustainability is increasingly regarded due to tightening emission regulations as well as demand from stakeholders, customers and business partners [19]. Furthermore, strong orientation toward environmental sustainability within a company could improve that company's overall competitiveness [20]. Additionally, disregard or inaction concerning environmental sustainability could impose unexpected costs on a company [21], and it could be in companies' best interest to implement proactive measures to mitigate these possible costs [22].

Environmental sustainability and its possible competitiveness benefits in transportation can be enhanced by transport mode selection (e.g., utilizing multimodal transport chains with a larger share of less-emitting transport modes) and emphasizing collaboration within supply chains [23]. However, even though railways can be utilized to transport large amounts of freight conveniently and ecologically, road transports are often preferred due to higher mobility and flexibility [24]. Moreover, road transports are often used to support other modes of transport, such as in the pre- and post-haulage of railway transports [25]. To benefit from multimodal transports and the involvement of multiple actors within the chain, adequate intermodality and information sharing are required [26]. Additionally, an integrated supply chain requires a certain degree of trust between the involved actors [27]. Moreover, research by Ayoub and Abdallah [28] suggested that the benefits from flexibility are reached through innovativeness and responsiveness within a supply chain. It should also be kept in mind that multimodal transport chains are imposed to transaction cost every time a transport mode is changed [23,29,30]. Technologies such as Radio Frequency Identification (RFID) [31,32] and Wireless Sensor Networks (WSN) [32] could be utilized to support supply chains involving multiple separate actors through efficient information exchange. The tracking of goods offered by such technologies also enables monitoring of the reverse logistics, e.g., recycling of the packages [31].

To maintain the profitability of a business in an ever-changing world, deployed business models should evolve accordingly [33]. Furthermore, introduction of innovations to business should be conducted so that said innovations are woven into the company's business model [34]. In the case of incumbent companies, the existing business models and assets must be addressed in the business model renewal to avoid conflicts between new and established practices [35]. In addition to business model design, the implementation of the model is a significant factor in its profitability [36]. Boons and Lüdeke-Freund [37] claim a strong relationship between a company's business model and its innovation activities, where those activities enable not only innovative outputs but also business model renewal for competitive advantage. As established, the role of business model innovation is of high importance in economically sustainable business; however, often this process is not successful [38]. Due to this relationship of need and associated uncertainty, companies' dynamic capabilities towards renewing business models and their specific industry remain important factors in the business model innovation [39]. Especially in business model innovation aiming towards higher environmental sustainability, the surroundings of a company (e.g., industry, other actors and society) should be regarded in their business model through network orientation [40].

Hence, logistics service provision interconnects with environmental sustainability and business model innovation, mainly due to emerging trends in legislation and regulation as well as stakeholder demands. As stated, environmental sustainability is a growing issue in logistics [17,19] and in order for a company to reap the benefits from innovations, their business model should be designed in a manner allowing that [34,37,40]. This trajectory suggests that the logistics service providers should revise their business model design and include environmental sustainability as a factor of value for their offering. Furthermore, the increased competition and new entrants in railway traffic due to the changing business environment between Finland and Russia suggest that the companies related to this field should find ways to cope with the introduction of numerous new actors. Furthermore, environmentally sustainable business models, such as those based on circular economy practices, require extensive cooperation between separate actors [41].

## 3. Materials and Methods

This article combines a research from the latter part of 2018 (see [42]) with newer empirical study from 2019. The former research investigated the possible new trends in business models associated with manufacturing and transportation companies situated in Southeast Finland. Furthermore, the interplay of relevant innovations and new business models in the given context were studied. The previous study was conducted as a case study (e.g., [43,44]) with 10 semi-structured interviews of Finnish and Russian transportation professionals and experts, and a survey for manufacturing and transportation

companies in Southeast Finland. A qualitative approach to studying emerging markets was used, as proposed by Guillotin [45]. The surveyed companies were mostly small- or medium-sized companies (SMEs) handling raw materials or low manufacturing value products. Findings from the previous study were used as a baseline to investigate the diffusion of the most promising business models and innovations since the last period of investigation. From the point of view of the studied companies, contemporary relevant and feasible innovations lie in the sphere of environmental sustainability. In transportation, these include technologies such as alternative fuels (e.g., LNG and electricity), and operations improvement (e.g., transport mode selection and multimodal transport chains).

In order to examine the validity of previous findings and to further study the development of the transportation industry in Southeast Finland, a newer survey study conducted in the autumn of 2019 concerning road transportation between Finland and Russia was conducted. This study of international road transportation was executed in the form of a web-based survey, which was distributed to 919 companies operating in the field of transportation, logistics and forwarding situated in South Finland alongside the highway E18 (European Road, a main serving road between Finnish and Russian trade). This research scope covers regions from Finland's west coast to its capital area and furthermore to its eastern border with Russia (starting in the west from the greater Turku region, and continuing to the capital region of Helsinki and from there onwards to the eastern border, ending at the Vaalimaa border-crossing point). A sample size of 56 recipients responded to the survey, setting the response rate for this survey at approximately 6%. This survey was more successful in terms of response rate, possibly due to its more specific scope and the shorter time required to fill the survey. The conducted survey focused on the past performance and future projections of local, national, international and transit road traffic in Finland, as well as perception towards alternative fuels in road transports and the usage of multimodal transport chains.

In addition, secondary data from an open-access database on road traffic near border crossing points between these countries, provided by the Finnish Transport Infrastructure Agency [46], was used to examine road freight traffic between Finland and Russia on a macroscale. A map with these border crossing points can be found in Appendix A. The described approaches were used jointly to gain a deeper understanding through triangulation on the complex problem setting established by the research questions [47].

## 4. Case Study Findings

The findings are presented in such a way that the results from the previous study are presented briefly first in order to prime the reader to the context of this case study. Thereafter, the results from the succeeding study are examined against the backdrop of the information discovered in the preceding research. While the first study had a broader scope in terms of the studied industries and factors in their respective environment, the results from this research indicate the need to focus on the transportation operations. The evidence pointed out that the companies in Southeast Finland focus their innovation activities to promote environmental sustainability on the operations, and transportation practices were the most obvious target for optimization.

### 4.1. Previous Findings from Manufacturing and Transportation Companies

#### 4.1.1. Semi-Structured Interview Results

The research began with semi-structured interviews to map the present situation within the studied region and to let the involved experts and professionals share their vision about the current situation as well as the direction of future development. The central topics that emerged during the interviews are summarized in Table 1. The trends among the interviewees seem to focus on the growing demand of subcontractors and competition, as well as the fragmentation of the market into a multitude of separate actors and a higher number of smaller customers than before. The innovation activities seem to emphasize solutions to environmental challenges and ways to improve collaboration between

different actors via communication channels and information flows. As this research focuses on the Finnish–Russian international business environment, the Eurasian Land Bridge, and more specifically the railway connection from China through Kazakhstan and Russia to Finland, plays an important role (other routes to China also exist through Russia and Mongolia, but the Kazakh route is currently used). Regarding this, the interviewees had recognized growing volumes of international freight traffic on railways between Finland and Russia. However, Finnish logistics operators have experienced a decrease in their internal operations on the Russian side, but they remained optimistic regarding future investments toward operations in Commonwealth of Independent States (CIS) member countries. Likewise, interviewees from Russia saw potential in the railway connection of Europe and the Far East through the Eurasian Land Bridge. In addition, the importance of the Northern Sea Route seems to hold more significance amongst Russian interviewees. While the overall trajectory of affairs seems similar on the Finnish and Russian sides, the development in Russia aims to capitalize on the growth potential of the logistics industry by shifting more of the handling of the freight flows to local actors instead of foreign actors, i.e., seeking lower use of transit countries in imports and exports. This development has played a role in diminishing freight traffic on roads between Finland and Russia.

**Table 1.** Overview of the emerged topics during the interviews (modified from [42]).

| | Finnish Interviewees | Russian Interviewees |
|---|---|---|
| General remarks on the international logistics industry | Share of railway freight is growing between Finland and Russia.<br>Railway connection from Finland to China has challenges in the intermediary border crossings.<br>International operations target CIS countries, Mongolia and China.<br>The Russian market has fragmented from a few large customers to numerous smaller ones.<br>The Imatra–Svetogorsk border crossing point could be used to relieve pressure from other points. | The Northern Sea Route alongside supporting infrastructure is being developed.<br>High importance of a Russian railway corridor between West Russia and the Far East as an alternative to the conventional sea routes.<br>Containerization rate is still low in comparison to Europe.<br>Balance of imports and exports is offset by decreasing imports and stagnant exports.<br>Local ports are increasingly favored over transit countries. |
| National logistics infrastructure and competition | Ongoing and planned development of infrastructure.<br>Some disagreements on the emphasis of development.<br>There is demand for new entrants in the railway industry to create more flexible supply networks.<br>Competition on railways is fierce; few actors handling bulk material are realistically competing.<br>Role of the state in stimulating competition on railways.<br>Subcontracting and other supporting services. | Infrastructure in Central and East Russia is not optimal, but it is being developed.<br>Intense competition.<br>Russian railways (RZD) remains as a focal actor in the industry.<br>Political and economic uncertainty is a challenge, but there is development and growth potential. |
| Innovation in the logistics industry | Research and development activities emphasize environmental sustainability.<br>Blockchain technology could improve communication between separate actors, information exchange and tracking of shipments, and cut costs by reducing unnecessary slack within the logistics operations. | Common platform to unify separate actors is being developed.<br>Academy and businesses show interest toward Blockchain technology.<br>The environmental sustainability of logistics industry is not being actively developed. |

Moreover, the railway infrastructure in Southeast Finland is seeing investments to its development, and at the same time the competition of railway traffic has become liberated, allowing other actors into the market in addition to the state-owned operator. However, some of the interviewees saw that the entry barriers to this field are too high for new entrants, expect for few operators specializing in certain types of bulk freight. At the same time, the incumbent actors on railways signaled their demand for new potential partners in the field to develop their network to fit customer demands more flexibly. Therefore, the Finnish interviewees called for state-level initiatives to stimulate the entry

of new actors into the market. The vision of interviewees from both sides of the border on relevant innovation in transportation related to the solutions seeking to manage and interact with a multitude of different actors in transportation industry. Additionally, on the Finnish side, interviewees saw the environmental sustainability of operations as a pressing issue due to the tightening regulations and rising demand for sustainability from customers as well as business partners. On the other hand, the industry on the Russian side seems to be in more dire need for solutions to communicate efficiently with the growing number of actors in the transportation field.

### 4.1.2. Survey Results

In the 2018 survey, approximately half of the respondent companies had engaged in international business. Approximately a quarter (23.1%) of the surveyed companies focused on exporting, whereas approximately 11.5% were focused on importing. A group of 7.7% were both exporting and importing and the rest (7.7%) reported doing other international business. As can be seen in Figure 1, a share of 42.9% of the companies with international operations had those within the EU. The popularity of the EU is most probably due to low barriers for the movement of goods, people and capital within the region. Since the geographical location of the studied region lies in the border area of Finland and Russia, the next largest target for foreign operations was Russia and other CIS countries. The markets were targeted by a group of companies with the shares of 23.8% and 4.8% for Russia and other CIS countries, respectively. Interestingly, despite the historical, cultural and geographical proximity of other Nordic countries, fewer companies indicated them as their target market. Lastly, some of the studied companies also had operations towards China and other Far East regions, but other locations were not mentioned by the respondents. The remaining half of the respondents without foreign operations signaled no interest to establish them.

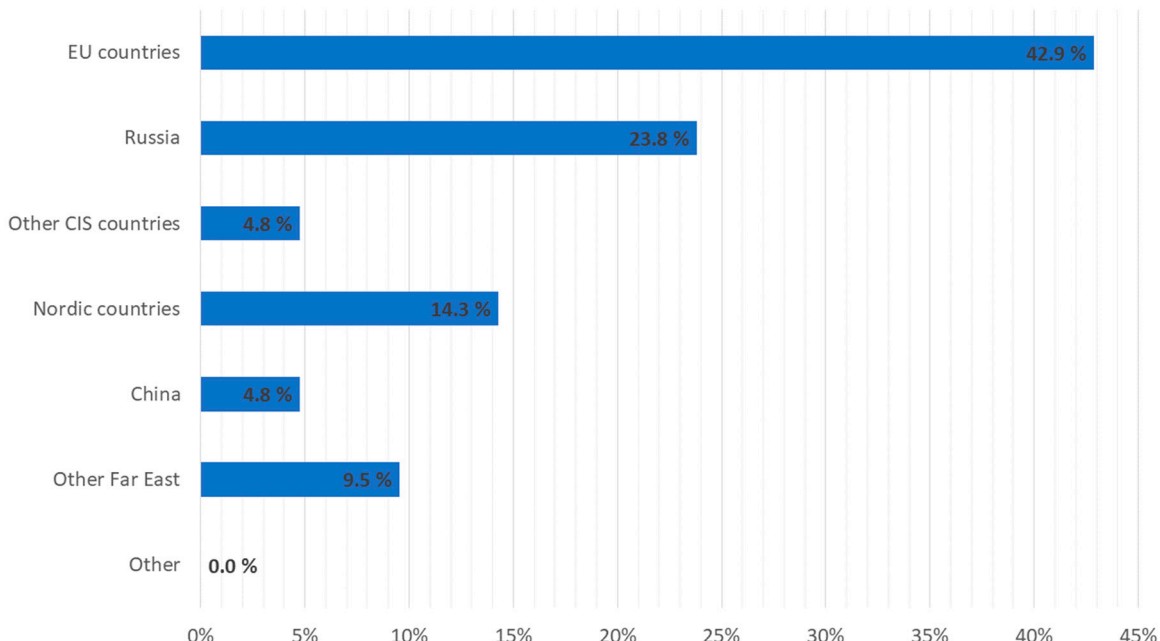

**Figure 1.** Target countries for international operations of the surveyed companies.

The previous survey was set to investigate the effect of changes in legislation and regulation to the business of the surveyed companies. While many of the presented changes were not regarded as impactful, the carbon dioxide ($CO_2$) mitigation directive was seen to have a relatively distinct effect by the companies, as illustrated in Figure 2. This is not very surprising, as the transportation sector has a significant stake in generated $CO_2$ emissions globally. While 42.3% did not grade the effect, and 15.4% saw no impact by the directive to their business, the rest of the group valued some effect for the

EU's $CO_2$ mitigation strategy. Since the trajectory of $CO_2$ emissions originated from transportation, the reduction of those emissions requires most probably some radical changes in the used vehicles or how they are operated (e.g., alternative fuels with lower $CO_2$ emissions). The impact is difficult to estimate, but transportation companies will have to renew their fleet from its current composition.

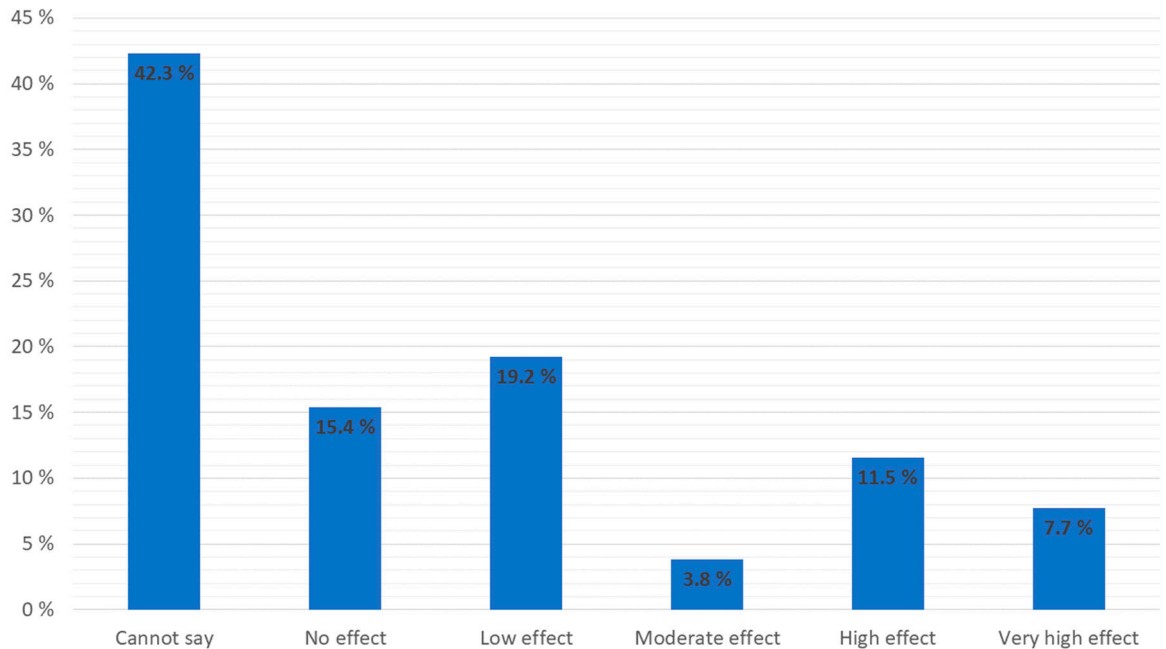

**Figure 2.** Effect of the EU's carbon dioxide emission mitigation directive on the surveyed companies' activities (Likert scale; 0 = cannot say; 1 = no effect; 2 = low effect; 3 = moderate effect; 4 = high effect; 5 = very high effect).

The transportation and manufacturing industries follow the same development as other sectors of the economy, where service provision has taken a significant share of all transactions in the market. This phenomenon is portrayed in the survey, since approximately half (46.2%) of the surveyed companies indicated that they require subcontracting services to support their business. At the same time, a large share of the respondents (57.7%) was offering subcontracting services to other companies. This overlapping of service provision and consumption can be seen in Figure 3. The survey allowed the respondents to write free-form comments about subcontracting, and some of the responses pinpointed that companies offer subcontracting back and forth to each other, whenever the need arises. This type of behavior has been observed by Hedenstierna et al. [48] in 3-D printing operations in Europe. Moreover, some of the interviewees indicated their need for more subcontractors to help them serve their customers in a more flexible manner. Additionally, the Russian interviewees indicated the growing importance of service provision in the logistics sector in Russia, likewise pointed out in the research by Yakunina [49].

Based on the interviews conducted before the distribution of the survey, the most relevant innovations according to the interviewees were studied with the help of the surveyed companies. These were blockchain, the Internet of Things (IoT), Artificial Intelligence (AI), LNG, catalyst converters for emissions in sea vessels, bio-economy or the utilization of renewable energy sources and circular economy. The surveyed companies were asked to rate their interest towards applying the mentioned innovations in their respective business practices. In addition, the respondents were asked to indicate if they have already implemented some of the mentioned innovations, or if they plan to do so.

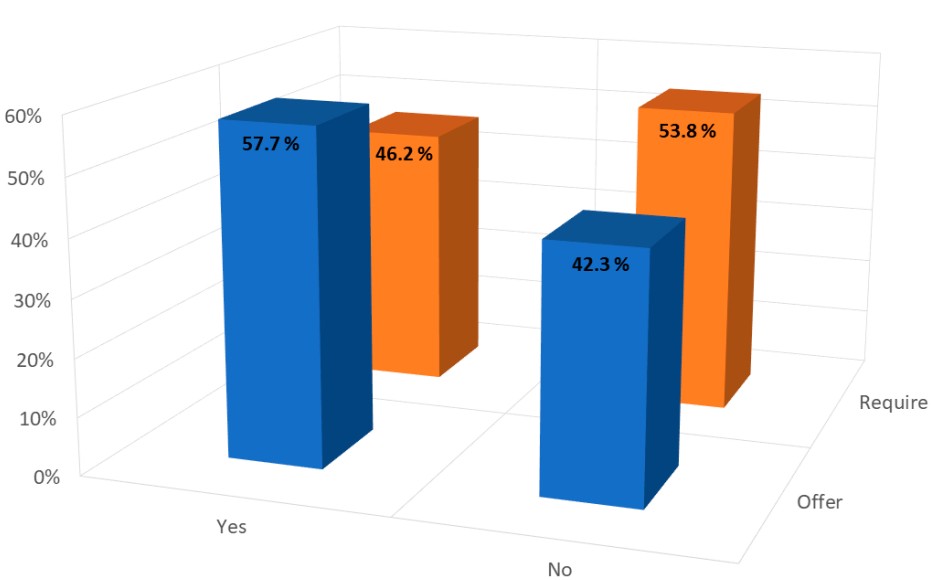

**Figure 3.** Supply and demand for subcontracting by the surveyed companies.

As illustrated in Figure 4, the innovations related to higher environmental sustainability scored higher grades from the respondents. Exceptions were LNG as fuel and catalyst converters, which are more relevant to maritime transports (although in fact LNG is being experimented with in road transports [50–52]). While shipment monitoring was revealed as a topic of high interest for transportation companies during the interviews, technologies such as blockchain and IoT received a low score on interest during the survey. Blockchain was deemed not interesting for the surveyed companies, possibly due to the debatable maturity of the solutions based on this technology, despite it being already implemented in transportation activities by the Danish container logistics company Maersk [53]. Circular economy and bio-economy were perceived as the most interesting innovations from the ones presented to the surveyed companies, and respectively 42.3% and 23.1% of these companies were planning or had already implemented these in their business activities. From these results it is evident that the manufacturing and transportation companies see environmental sustainability as a main target for their innovation activities.

To conclude the survey, the respondents were asked to grade three distinct business models by their feasibility in the companies' respective business environment. These models are generalized examples of the visions for emerging business models by the interviewees. Moreover, the models strive to capture the benefits from legislation and regulations, the changing business environment of Southeast Finland, and new technology and innovations, which were studied during this research. Thus, the proposed business models for the respondents were innovative subcontracting-based, platform-based and blockchain-based models.

Firstly, the innovative subcontracting-based model in the context of this research refers to a model where the focal company offers subcontracting services on business sections that have not been externalized by the principal companies before. Furthermore, this type of model would enable lateral collaboration between companies to offer subcontracting services to each other, e.g., by order book smoothing as described by Hedenstierna et al. [48]. For example, the gradually liberated competition on railway traffic could offer opportunities for this type of business activities. Secondly, the platform-based model involves a situation where the existing markets operate within a digital platform. The initiative for companies to join this community would be the convenience through a streamlined process of sourcing providers and identifying customers. In addition, a platform could stimulate competition through a less formal contract structure and more transparent tendering. Additionally, as the studied market is more fragmented in terms of number of separate actors; according

to the interviewees, a platform could act as a tool to navigate this increasingly complex network. Lastly, the blockchain-based model in this research can be understood similarly to the platform-based one, but the point here is not focusing on digitalizing the marketplace. This model seeks to reinforce the existing networks by allowing a more efficient exchange of information between partners, by making the transactions more transparent by involving the whole supply chain in the information exchange and by verifying the transactions within a supply chain by the participants of said chain.

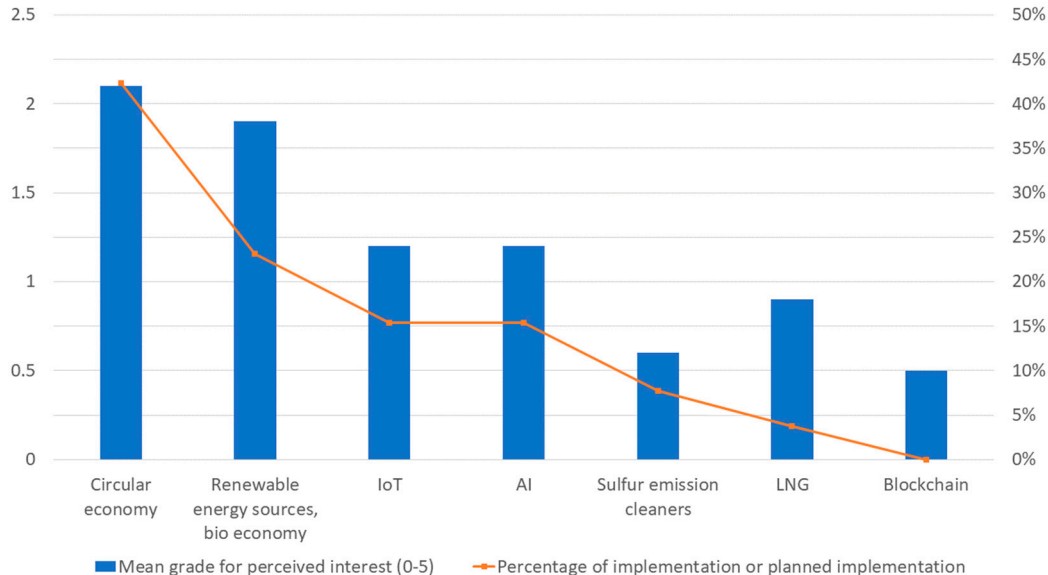

**Figure 4.** Perceived interest and implementation rate of the studied innovations (modified from [42]; grading for interest was done on a Likert scale; 0 = cannot say; 1 = no interest; 2 = low interest; 3 = moderate interest; 4 = high interest; 5 = very high interest).

The different proposed business models and their perceived feasibility are combined in Figure 5. As was established in the earlier results, the need for subcontracting services in the studied region is significant. This can be seen manifested in the perceived feasibility grade for the innovative subcontracting-based model, which had a feasibility rating of "very high" for 15.4%, "high" for 7.7%, and "moderate" for 19.2% of the respondents. The other two proposed models received a lower feasibility rating, and the platform-based model was seen as slightly more feasible. The modest success of those models can possibly be explained by the low maturity of the required technology in the given context of manufacturing and transportation SMEs in Southeast Finland. A more significant factor in the low perceived feasibility for these models probably is that they require extended trust between the separate actors introduced into the network [54], which is not typical in industries with fierce competition as the ones studied. The interviewees also recognized this challenge in deploying the mentioned business models.

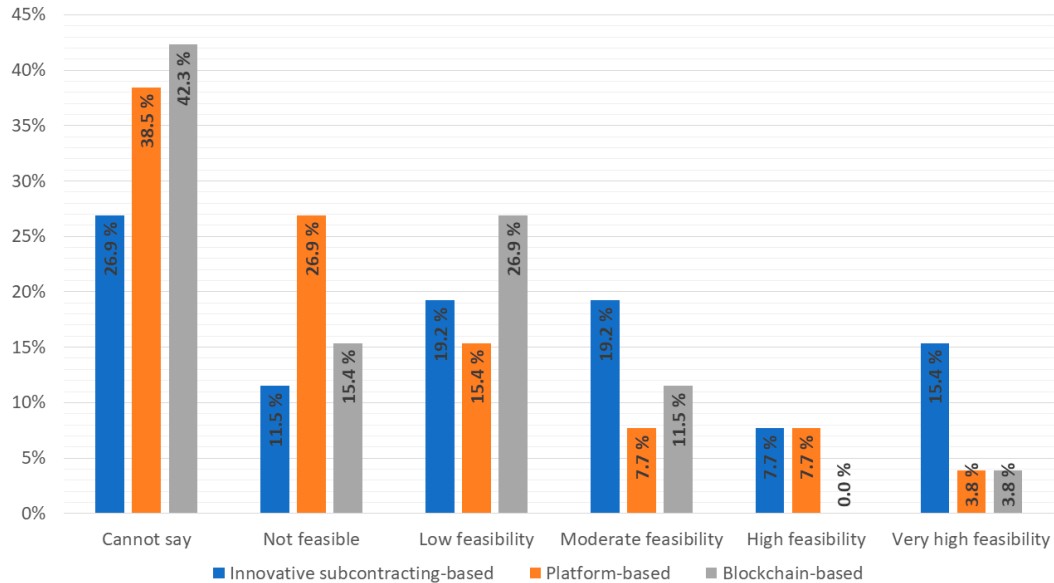

**Figure 5.** Feasibility of the proposed business models (Likert scale; 0 = cannot say; 1 = not feasible; 2 = low feasibility; 3 = moderate feasibility; 4 = high feasibility; 5 = very high feasibility).

*4.2. Findings from Transportation Companies of South Finland through A Second Survey*

As established during the previous study, environmental sustainability emerged as the driving factor for business model renewal as well as research and development in the manufacturing and transportation industries of Southeast Finland. The described legislation and regulation changes concerning emissions are currently gradually coming into effect, and their impact on energy intensive industries such as transportation are yet to be observed. Therefore, the succeeding study is scoped to emphasize environmental sustainability and transportation companies. A survey on road transports and the usage of highway E18 in Finland, targeting transportation, logistics and forwarding companies, was conducted in the autumn of 2019. The results of this survey interconnect with the topic and results of the previous survey, although it is focused on the road transportation mode. Based on this survey, the barriers of internationalization still exist and are perceived as strong for Finnish transportation companies considering Russia as a target market. A summary of the topics that emerged from the free-form comments about road freight traffic between these countries by the survey respondents can be found in Table 2.

**Table 2.** Topics that emerged from the free-form comments on the road freight traffic between Finland and Russia by the surveyed companies.

| | |
|---|---|
| International road transport has lower volumes of cargo than before. | The renewed highway E18 on the Finnish side is a safe and working road. |
| The border formalities have become stricter and therefore take more time, which disrupts the traffic flows. | Transport business between Finland and Russia is volatile, thus not very appealing for Finnish companies. |
| Road use taxation sets challenges for international operations. | Demand for services targeted to the professional users of road E18. |
| The Russian side of E18 (Scandinavia road) only has one lane near the border; 2–3 lanes would allow for a smoother flow of road traffic and enhance safety. | Russian companies handle most of the international road transports between Finland and Russia. |

The operators based in Finland see the Russian market as uncertain and volatile, i.e., the perceived risks are higher than the perceived benefits. The same observation was made in the interview phase, and interviewees from the Russian side share the view of market volatility to a certain degree. One of the more glaring barriers to international road transportation are the strict border formalities, which disrupt the fluency of the traffic flows. Reportedly, another factor in the undesirability of international

road transport activities is the inadequateness of the road infrastructure; onwards from the Finnish border station Vaalimaa, the E18 European Road has only one lane, which not only undermines the traffic flow, but also decreases the safety on a busy road. From the Finnish perspective, it must also be acknowledged that the pricing for transports originating from Russia are lower than those from Finland. The majority of the road transports originating from (or transiting through) Finland headed to Russia are undertaken by Russian operators. This shift of emphasis in the responsible companies from Finnish actors to Russian ones could arguably be one of the factors explaining the relatively unenthusiastic view on the prospects of international road transports from the Finnish side. This, connected to the lower import activity of Russia (also concluded from the interviews), could be used to explain why business opportunities in international road transports between these two countries are not exceptionally flourishing.

Furthermore, the development of road transportation in Southeast Finland is seen by the respondents as stagnating. As shown in Figures 6 and 7, the road transports between Finland and Russia through main border crossing points between these countries have been constantly decreasing from 2010. A main contributor towards this development is the disappearance of transit freight traffic via road between Finland and Russia [55]. While in recent years the road transport volumes are nowhere near the amounts of 2010, traffic seems to be returning slowly to Vaalimaa and Nuijamaa border stations, whereas activity at the Imatra border crossing point continues to decrease in both directions. As has come up in the interview phase of this research, material flows originating from and arriving to Russia are increasingly shipped from and to local seaports. Therefore, once-active transit traffic through Finland (ships arriving to Finland and the goods being transported to Russia via road) can be seen decreasing drastically.

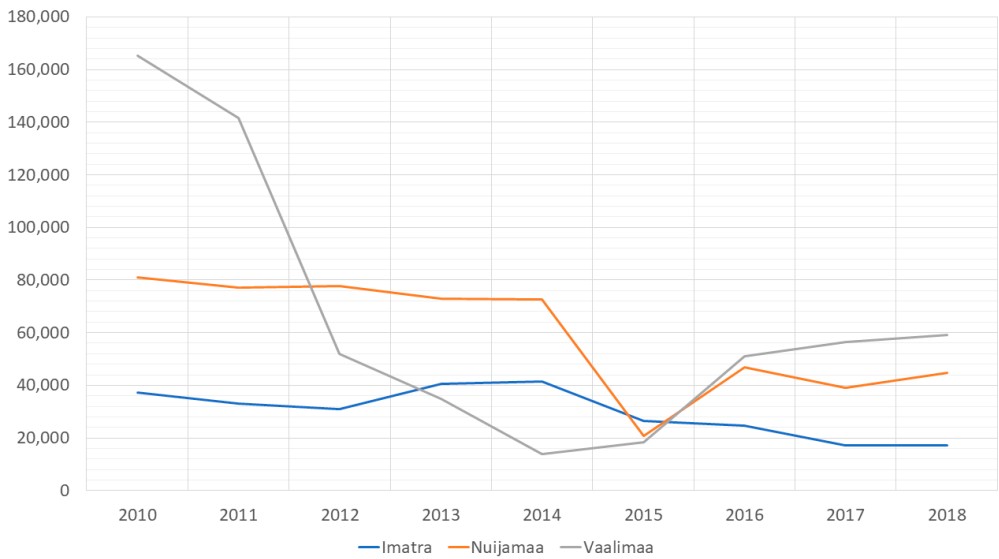

**Figure 6.** Road freight traffic (number of vehicles) from Finland to Russia [46].

As can be seen in Figure 8, road transport practitioners' perceived feasibility of alternative fuels remains on the low end—similarly to the situation in the earlier survey. While it could be spotted that some companies are actively getting ready to implement a higher biocomponent share in diesel and even LNG in their road transport activities, most of the companies are hesitant to adapt these alternative fuels. Moreover, it seems that electric vehicles are not seen as relevant for transportation activities, i.e., companies do not see electric heavy-duty vehicles penetrating the market just yet. It is a peculiar situation, since alternative fuels require upgraded infrastructure to offer operation range for vehicles running on such fuels, such as service stations with natural gas pumps or electric vehicle charging spots, but on the other hand road traffic support service providers do not experience enough demand to invest in these upgrades. However, it should be noted that when addressing

total carbon dioxide emissions originating from transportation, thorough consideration of the large picture regarding environmental sustainability is required. For example, large-scale investments in the electric vehicle fleet could in fact increase total emissions from transportation, since a majority of electricity is still produced with fossil fuels [6]. This challenge acts as a proof for the complexity of sustainability challenges.

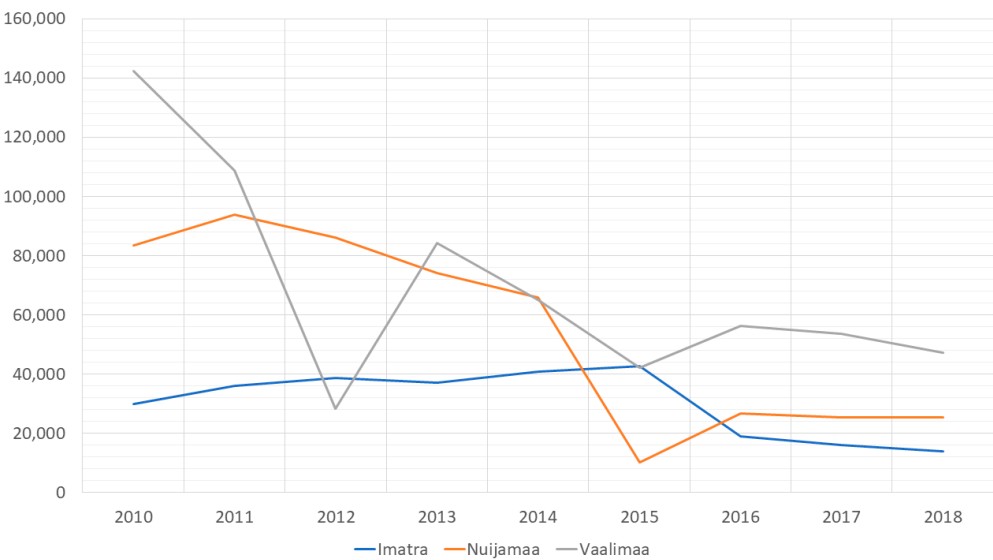

**Figure 7.** Road freight traffic (number of vehicles) from Russia to Finland [46].

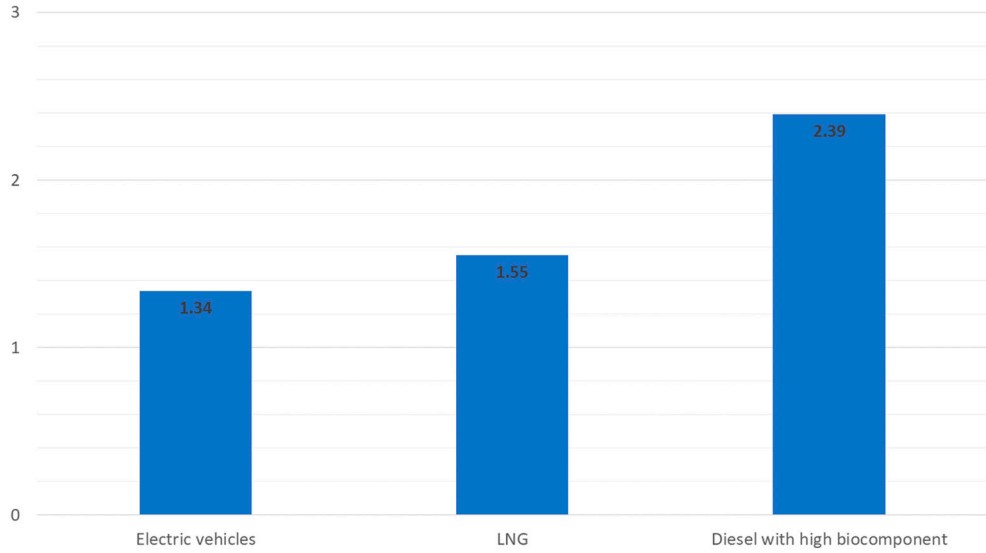

**Figure 8.** Average grades for feasibility of alternative fuels in road transportation (Likert scale; 0 = cannot say; 1 = not feasible; 2 = low feasibility; 3 = moderate feasibility; 4 = high feasibility; 5 = very high feasibility).

The existence of internal warehousing activities seems to drive the importance of road transports and related infrastructure for a company. A share of 36% respondents from the surveyed companies had internal warehousing activities, and the locations of these warehouses were mostly on the eastern sections of highway E18 (78% of warehouses were either in Kotka, Vantaa or Kouvola). Respondents from this region indicated that highway E18 is important for their operations, more so than those from the western regions (82.4% of companies with warehouses in the eastern parts signaled the importance

of E18, whereas the share was 40% for those with warehouses more to the west). This correlation could arise from the fact that most of the warehouses are only accessible by road. Only a certain number of companies overall benefit from the strategic positioning of warehouses, where different transport modes can be used efficiently, i.e., warehouses that are in the vicinity to railway tracks or marine ports. As emphasis on environmental sustainability is growing in transportation, environmental logistics service providers can capitalize on this apparent need of multimodal transport chains. Based on the premise and results of this research, sustainability, especially in terms of the environment, could emerge as an important factor in service provider tendering. Moreover, companies that are future-oriented could capitalize on this need with a state-of-the-art vehicle fleet, i.e., vehicles running on alternative fuels and high capacity transports. High capacity transports in this context refer to road trains, which are heavier and longer in comparison to conventional ones. In Finland these high capacity transports mean road trains that range from 76 to 100 tons in total weight, which have special permissions to operate on certain roads in the Finnish road network.

## 5. Discussion

One of the most influential legislation changes affecting the studied region is the liberation of competition on railway freight traffic, allowing new entrants to the market. The introduction of new entrants to railways could create demand for new road transport services to support the railway operations in the pre- and post-haulage phases of the transport. In addition, maritime traffic faces tightening global and local (Baltic Sea Emission Control Area) emission regulations, which chip away at this mode's competitiveness against other modes [9–12]. This development creates a need for road transport service providers, but in order to enable prolonged success these actors must meet adequate degree of environmental sustainability [20] to comply with the regulations of the examined region [7,8]. Furthermore, if more actors keep entering the market, a demand for managing the increasingly complex logistics network could arise. As discussed earlier in this article, state-of-the-art solutions exist to satisfy the described need for connecting a multitude of separate actors. However, the presented models seem to require extended trust between the involved actors, which seems to be a considerable barrier for their diffusion.

The international road freight traffic between Finland and Russia has been declining consistently during the last decade. While there are various policy and national strategy-level reasons (e.g., those leading to diminished transit traffic [55]), one reason is the increasing popularity of the railway connection between these countries. Wood, pulp and metal industry products are transported utilizing the railway infrastructure of the Eurasian Landbridge [56]. The railway mode also has the potential for increasing volumes for food (especially meat) transports from Northern Europe to China through Finland and CIS countries. Increased utilization of this route has also spawned businesses in Finland, who specialize in railway freight traffic between Finland and Russia.

The rapidly changing business environment implies the need for equally rapid business model renewal with corresponding managerial decision-making—a challenge for companies who wish to remain competitive [33–40]. Cued by the changing business environment and refined by the business model theory evolution, three general-level business models (innovative subcontracting-based, platform-based and blockchain-based) were proposed and their feasibility was graded by manufacturing and transportation companies of Southeast Finland. The models are designed to exploit the changes in the studied business environment as well as relevant emerging innovations (to acquire competitive advantage [37,39]) as indicated by the interviewees. The trust between the involved actors remains an inherent requirement with varying intensity for these models, which is also seen as a factor for success of supply chains [25,27]. While the interviewed experts voiced the need for efficient shipment tracking, the new technologies and business models enabling that need were not seen as feasible by the surveyed companies, possibly due to a lack of trust between actors in the examined region. The main emphasis of the studied companies' innovation activity lies in promoting their environmental sustainability. As proposed by Jasmi and Fernando [19], the studied companies' informed focus on environmental

sustainability is due to the stakeholder demands and tightening legislation and regulation towards emissions. Once again, the companies' business model should accompany this vision in order to create benefit from it [40]. However, the practice in the studied region seems to differentiate from the business model and innovation theory.

## 6. Conclusions

The changing regulations and environment regarding international business between Finland and Russia are moving the manufacturing and logistics industries towards a freer market with lower barriers for new entrants. At the same time, the requirements concerning the environmental impact of these business activities are becoming stricter. This development drives incumbent companies to re-design their respective business models, and acts as a cue for new actors to enter the market. In addition, the presented changes open avenues for achieving a competitive advantage with adapting the new set of rules over those who do not. Business models that are designed to capture value from growing number of actors in the market and environmentally sustainable business are likely to prosper amid the discussed changes.

New technology and innovation regarding information and communication technology (e.g., blockchain) enable more precise tracking of shipments [31,32] and more efficient communication between actors in a supply chain. If the required degree of trust can be established between the actors, these innovations can reinforce collaboration between them. Moreover, enhanced tracking of shipments and their origin enables the verification of sustainably sourced goods. Some of the studied companies seem to be ready for this kind of commitment, but a majority remains skeptic. Alternative fuels (e.g., LNG and electricity) for transportation could significantly lower the environmental impact of logistics operations. Furthermore, environmental sustainability can be improved via innovative business practices (e.g., circular economy). Means to reduce the negative impact to the environment were seen as more relevant among the studied companies.

The presented case of international transportation between Southeast Finland and Northwest Russia could be used as a reference point for studies concerning other countries that rely on railway transportation in import and export activities. Furthermore, the insights on the railway connection through the Eurasian Landbridge between the Far East and Northern Europe could benefit other regions, such as Central Europe. Especially when companies are looking to ease the environmental impact of their supply chain activities, the railway transport mode could offer means to reduce produced emissions. The studied technologies and innovations should also be considered in other regions. While the reception for the studied digital technologies was not overwhelmingly enthusiastic in the studied region, the case might be different in the context of other regions. The technologies and innovations focusing on reducing negative environmental impacts should be considered by companies looking to achieve a competitive advantage through environmentally sustainable business practices.

While this research is limited to a specific region, it is also limited by the general scope of manufacturing and logistics industries in that region. It would be recommendable to study business model and innovation theory closer to the practice in the future, e.g., through piloting in experimental environments with companies. Furthermore, it is important to study those theories in the context of high physical asset intensive (and low intellectual capital intensive) industries. Pieroni et al. [40] also call for more experimentation and learning from practice, and Poponi et al. [41] call for more generalizable models in environmentally sustainable business model design and innovation. The new situation, where bigger countries aim to produce more locally and import less, should deserve further research from the angle of international logistics companies. Lower demand for logistics services is not the only implication; services are also experiencing higher competition, and structures as well as transportation modes do seem to be changing.

**Author Contributions:** Conceptualization, O.L. and O.-P.H.; methodology, O.L. and O.-P.H.; validation, O.L. and O.-P.H.; formal analysis, O.L.; investigation, O.L.; resources, O.-P.H.; data curation, O.L.; writing—original draft preparation, O.L.; writing—review and editing, O.-P.H.; visualization, O.L.; supervision, O.-P.H.; project administration, O.-P.H.; funding acquisition, O.L. and O.-P.H. All authors have read and agreed to the published version of the manuscript.

**Funding:** The interviews and first survey were done as consulting work and the second survey was funded by the Southeast Finland–Russia Cross-Border Cooperation Program.

**Acknowledgments:** This article is based on a conference proceedings manuscript presented at the 23rd Cambridge International Manufacturing Symposium in September 2019 (Cambridge, UK). The previous research has been extended with new empiric and secondary data. We would like to thank the participants of this conference for their valuable comments and insights.

**Conflicts of Interest:** The authors declare no conflict of interest.

## Appendix A

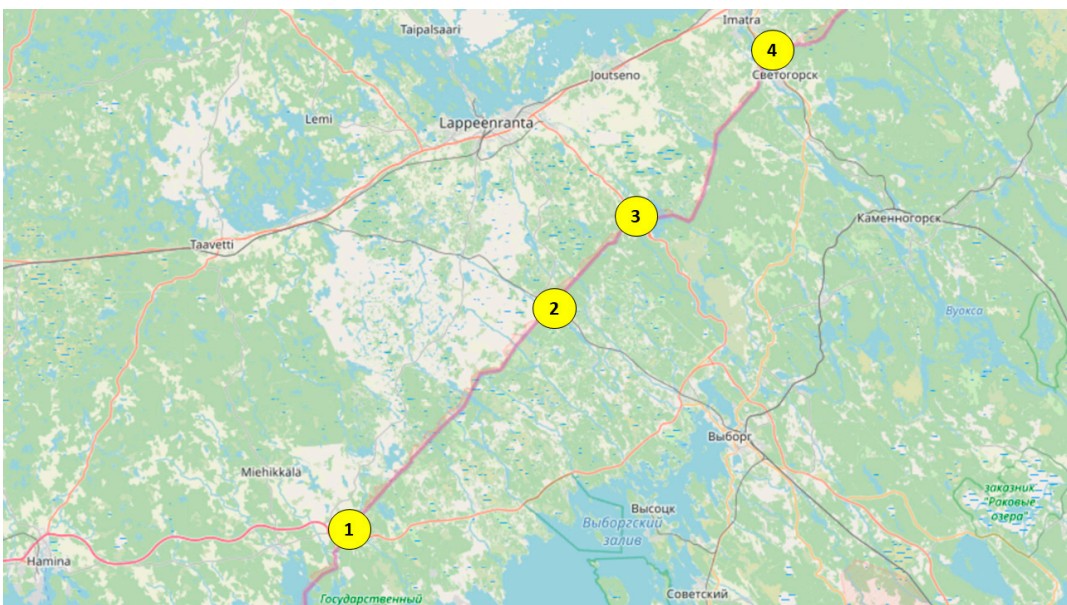

**Figure A1.** Border crossing points between Southeast Finland (left side) and Northwest Russia (right side; Leningrad region). Numbers in the figure correspond the name of the respective crossing point, as follows: 1 = Vaalimaa–Torfjanovka; 2 = Vainikkala–Buslovskaja (railway border-crossing point); 3 = Nuijamaa–Brusnitsnoje; 4 = Imatra–Svetogorsk [57].

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
