# Peer review of "Business Models Amid Changes in Regulation and Environment: The Case of Finland–Russia"

_sustainability, doi:10.3390/su12083393_

Round 1
Reviewer 1 Report
Dear authors,
Thank you for giving me opportunities to read your paper. I found your research quite important and interesting in terms of practical recommendation for the logistic issues. However, my major concern is about theoretical contribution of the paper which are not elaborated well in the current version of the paper. You can talk about the following contributions: What insights on the business model evaluation and innovation are needed? That may help you make more contributions and position your contributions better. My main suggestion is that you can link to more literature by discussing more relevant publication. You should consider, for example:
Makadock, R., Burton, R., Barney, J. (2018) A practical guide for making theory contribution in strategic management, Strategic Management Journal, 38(6),
Kim, S.K., Min, S. (2015) Business Model Innovation Performance: When does Adding a New Business Model Benefit an Incumbent, Strategic Entrepreneurship Journal, 9(1),
Bera Solis, H., Cassadesus-Masanell, R. Grief-Tatjē, (2015) Business Model Evaluation: Quantifying Walmart's Source of Advantage, Strategic Entrepreneurship Journal 9(1).
The last section should be called conclusion where you should summarize all you findings, their implication to researchers, future direction for research, limitation of the current study, etc.
In conclusion, I would like to thank the authors for a very interesting, and potentially important paper.
Hope these comments and suggestions can help your study.
Author
Author Response
Point-by-point response to reviewer comments (first revision)
Title: Business Models amid Changes in Regulation and Environment: Case Finland-Russia (Original title: Business Models as Regulation and Environment Changes: Case Finland-Russia)
Journal: Sustainability
ISSN: 2071-1050
Manuscript ID: sustainability-758621
The authors of the proposed manuscript would like to express their sincerest gratitude to the reviewers for their invaluable suggestions for improvements. These comments prompted major revisions for the manuscript and almost complete overhaul of the text. As a result, the quality of the manuscript was considerably increased. Revisions corresponding each of the reviewers’ comments are detailed in point-by-point manner in the following section. The changes in the text are highlighted with yellow color.
Language
Authors’ response:
As the reviewers raised rightful concerns about the language used in this manuscript, it has been largely rewritten.
Literature review
“It is well done. On p2 L92, your mention of trust is very important. I agree with your statement and also recommend to look at the importance of trust in the success of platforms (see Yoffie et al, HBR, 2019: https://hbr.org/2019/05/a-study-of-more-than-250-platforms-reveals-why-most-fail).” – Reviewer 2
“However, my major concern is about theoretical contribution of the paper which are not elaborated well in the current version of the paper. You can talk about the following contributions: What insights on the business model evaluation and innovation are needed? That may help you make more contributions and position your contributions better. My main suggestion is that you can link to more literature by discussing more relevant publication. You should consider, for example:
Makadock, R., Burton, R., Barney, J. (2018) A practical guide for making theory contribution in strategic management, Strategic Management Journal, 38(6),
Kim, S.K., Min, S. (2015) Business Model Innovation Performance: When does Adding a New Business Model Benefit an Incumbent, Strategic Entrepreneurship Journal, 9(1),
Bera Solis, H., Cassadesus-Masanell, R. Grief-TatjÄ“, (2015) Business Model Evaluation: Quantifying Walmart's Source of Advantage, Strategic Entrepreneurship Journal 9(1).” – Reviewer 1
Authors’ response:
The article by Yoffie et al. is indeed relevant for this research. It has been regarded in the text and added to the references.
The mentioned articles (Makadok et al., 2018; Kim and Min, 2015; Brea-Solis et al., 2015) were helpful in enhancing the theoretical contribution of the research. Also, the research of Kim and Min (2015) and Brea-Solis et al. (2015) have been regarded in the literature review and the results have been reflected with those studies in mind. The mentioned studies have been also added to the references.
Conclusions
“The last section should be called conclusion where you should summarize all your findings, their implication to researchers, future direction for research, limitation of the current study, etc.” – Reviewer 1
Authors’ response:
This remark about the structure of the text has been regarded, the title for the section has been changed correspondingly (as “Conclusions”). The contents of the Conclusions section have been revised accordingly.

Reviewer 2 Report
Dear Authors/Colleagues:
This manuscript is fairly well written. However, I wonder about the originality/novelty value, and applicability of this case. In addition, I think that your title may need to be re-phrased as it is currently unclear to me. Do you mean: "new business models amid regulation and environment changes: the case of the Finland-Russia border?
Abstract:
Your style could also be improved by adding "a" and "the" in the text where needed to improve the readability of your manuscript (e.g., Abstract, L12 before case, L15 before growing, and p2 L46 before general). Also on p1 L 23, I suggest replacing regulation by regulatory. As you know, your abstract should be as perfect as possible and should also include some specific dates about the changes, the data collected, and results (new business models). In terms of findings, on p1 L21-22, you mentioned that volumes have not improved (i.e., increased?) but state on p5 L202 (Table 1) that "traffic of goods is growing between Finland and Russia" This contradiction needs to be rectified. On p3 L 98, I would add "the" in front of world. On p4 L164, I suggest replacing "second-hand" with "secondary" in front of data. I trust that this is what you meant.
Methodology:
You are leveraging two other studies which can increase the value and credibility of your findings, as well as the richness of your manuscript/paper. Your point on qualitative research in emerging markets (p3 L121-122) is very well taken. It is consistent with the methodology recommended in this MDPI 2019 paper, downloaded approx 1500 times in 15 months, that you may want to cite since I did not see any MDPI references in your list: https://www.mdpi.com/1911-8074/12/1/8
Additionally and in my opinion, some of your RQs seem better suited for a qualitative approach (impact of changes and measurement thereof for RQ1). Maybe you could rephrase RQ1 to avoid any unintentional confusion. More importantly, the relatively small number of interviews (10) with sparse data collection and analysis details failed to convince me that you reached saturation after your coding. Personally, I would have expected at least 15 interviews of at least 30 minutes and some details about the companies and informants they represent. As you probably know, your data should be systemically collected, triangulated (see, e.g., Burton and Obel, 2011) and your process entirely transparent, as well as detailed, in order to increase the credibility of your research. I suggest adding a timeline associated with the regulatory changes, timing of interviews, and the emergence of new business models in the context of the changes in the environment.
Literature review:
It is well done. On p2 L92, your mention of trust is very important. I agree with your statement and also recommend to look at the importance of trust in the success of platforms (see Yoffie et al, HBR, 2019: https://hbr.org/2019/05/a-study-of-more-than-250-platforms-reveals-why-most-fail).
Findings:
In Table 1, you mentioned the Imatra-Svetogorsk border crossing. It is unclear where this is exactly. A map as an appendix would be helpful to your reader.
Additionally, the connection between the changes and the evolution of new business models (how many?, how big are those companies?, etc.) is not clear to me, in regards to timing and conceptualization of your research model.
In conclusion, this manuscript relies on rich data (different surveys and perspectives) and could be improved with some revisions. I hope that this review is helpful to you and wish you continued success.
Regards,
Anonymous Reviewer
Author Response
Point-by-point response to reviewer comments (first revision)
Title: Business Models amid Changes in Regulation and Environment: Case Finland-Russia (Original title: Business Models as Regulation and Environment Changes: Case Finland-Russia)
Journal: Sustainability
ISSN: 2071-1050
Manuscript ID: sustainability-758621
The authors of the proposed manuscript would like to express their sincerest gratitude to the reviewers for their invaluable suggestions for improvements. These comments prompted major revisions for the manuscript and almost complete overhaul of the text. As a result, the quality of the manuscript was considerably increased. Revisions corresponding each of the reviewers’ comments are detailed in point-by-point manner in the following section. The changes in the text are highlighted with yellow color.
Language
Authors’ response:
As the reviewers raised rightful concerns about the language used in this manuscript, it has been largely rewritten.
Title
“I think that your title may need to be re-phrased as it is currently unclear to me. Do you mean: "new business models amid regulation and environment changes: the case of the Finland-Russia border?” – Reviewer 2
Authors’ response:
We agree that the title was a little ambiguous and made changes according to the suggestion. The study concerns the impact of regulatory and business environment changes to the business models used by the affected companies. Therefore, the new title stands as “Business Models amid Changes in Regulation and Environment: Case Finland-Russia”.
Abstract
“As you know, your abstract should be as perfect as possible and should also include some specific dates about the changes, the data collected, and results (new business models). In terms of findings, on p1 L21-22, you mentioned that volumes have not improved (i.e., increased?) but state on p5 L202 (Table 1) that "traffic of goods is growing between Finland and Russia" This contradiction needs to be rectified.” – Reviewer 2
Authors’ response:
The abstract was revised according to the above remarks to communicate the research more clearly to the reader. Furthermore, the mentioned contradiction has been correct. Table 1 referenced to the growth of share of railway freight in overall transportations. As a result, Table 1 was also remade alongside the abstract.
Research Questions
“Some of your RQs seem better suited for a qualitative approach (impact of changes and measurement thereof for RQ1). Maybe you could rephrase RQ1 to avoid any unintentional confusion.” – Reviewer 2
Authors’ response:
The first research question has been rephrased to represent the conducted research more clearly.
Methodology
“You are leveraging two other studies which can increase the value and credibility of your findings, as well as the richness of your manuscript/paper. Your point on qualitative research in emerging markets (p3 L121-122) is very well taken. It is consistent with the methodology recommended in this MDPI 2019 paper, downloaded approx 1500 times in 15 months, that you may want to cite since I did not see any MDPI references in your list: https://www.mdpi.com/1911-8074/12/1/8” – Reviewer 2
“More importantly, the relatively small number of interviews (10) with sparse data collection and analysis details failed to convince me that you reached saturation after your coding. Personally, I would have expected at least 15 interviews of at least 30 minutes and some details about the companies and informants they represent. As you probably know, your data should be systemically collected, triangulated (see, e.g., Burton and Obel, 2011) and your process entirely transparent, as well as detailed, in order to increase the credibility of your research. I suggest adding a timeline associated with the regulatory changes, timing of interviews, and the emergence of new business models in the context of the changes in the environment.” – Reviewer 2
Authors’ response:
The concern about methodology was well received. It prompted the authors to completely rewrite the “Materials and Methods” section to be more explicit of the used methodology, with the above remarks in mind. The article from Guillotin (2018) was beneficial read and it justifies the approach used in this study. That article has been also added to the references. Also, as the research employed triangulation of multiple different methods, it has been communicated clearly to the reader, with appropriate reference to Burton and Obel (2011).
Findings
“In Table 1, you mentioned the Imatra-Svetogorsk border crossing. It is unclear where this is exactly. A map as an appendix would be helpful to your reader.” – Reviewer 2
“The connection between the changes and the evolution of new business models (how many?, how big are those companies?, etc.) is not clear to me, in regards to timing and conceptualization of your research model.” – Reviewer 2
Authors’ response:
A map with the four border crossing points between South-East Finland and North-West Russia has been added to the Appendix A, as it is helpful for readers not familiar with this region. The evolution of business models in the context of this case study (company sizes and number of companies) has been communicated more clearly in the revised version. Also, in the latter parts of the article, the regulatory and business environment changes have been connected to the study findings more explicitly. Table 2 was changed into a diagram to enhance its informativity.
Originality
“I wonder about the originality/novelty value, and applicability of this case.” – Reviewer 2
Authors’ response:
This remark cued the revision of parts outside the specific comments (i.e., Introduction, most of Literature review, and Findings). The authors of this manuscript hope that the made changes as detailed in this letter are satisfactory to communicate the originality value of the research. Also, as the research is now communicated in a clearer fashion, it is hopefully more applicable.

Round 2
Reviewer 1 Report
Dear Authors, thank you very much for revising the paper. You did a good job in improving the paper. It is now better to read and has a better structure to follow.
Nevertheless, there is some more potential to improve the paper. After the case studies and prior ro the conclusion I would add a "discussion" section. Also things like "lesson learned". Is this transferable to other sectors and regions? I would also recommend that you should refer to the answers of research questions. I would answer it shortly.
Regarding "conclusion" I 'd rather like a clear answer on the third research question, namely, how innovations enabling these business models, what we know now more after your research, how your research contribute to current scientific discussion on the business model changes, what is a limitation of this research, and your future work. See for example: Parida et al., (2019), Reviewing Literature on Digitalization, Business Model Innovation, and Sustainable Industry: Past achievements and Future Promises. Sustainability.
Reviewer
Author Response
Point-by-point response to reviewer comments (second revision)
Title: Business Models amid Changes in Regulation and Environment: Case Finland-Russia (Original title: Business Models as Regulation and Environment Changes: Case Finland-Russia)
Journal: Sustainability
ISSN: 2071-1050
Manuscript ID: sustainability-758621
The authors of the once revised manuscript would like to thank the reviewer for the additional suggestions. These comments further helped to improve the quality of the manuscript. Revisions corresponding to the reviewer’s comments are detailed in a point-by-point manner in the following section. The changes in the text are highlighted with yellow color.
Language
Authors’ response:
Upon revising the manuscript, some grammar and typographical mistakes were spotted. These have been corrected and highlighted in the text.
Discussion
“Nevertheless, there is some more potential to improve the paper. After the case studies and prior to the conclusion I would add a "discussion" section. Also things like "lesson learned". Is this transferable to other sectors and regions? I would also recommend that you should refer to the answers of research questions. I would answer it shortly.” -Reviewer 1
Authors’ response:
The recommendation of adding discussion section was received well. It has been added after the Findings and before Conclusions. Since Conclusions section contained text that reflects the findings with the research questions and the studied literature and theory, it was moved to Discussion section. Also, further implications based on the findings and discussion, transferability, and answers to the research questions have been added. However, they seem to fit more fluently to Conclusions section. Hence, these parts have been revised and added in that section.
Conclusions
“Regarding "conclusion" I 'd rather like a clear answer on the third research question, namely, how innovations enabling these business models, what we know now more after your research, how your research contribute to current scientific discussion on the business model changes, what is a limitation of this research, and your future work. See for example: Parida et al., (2019), Reviewing Literature on Digitalization, Business Model Innovation, and Sustainable Industry: Past achievements and Future Promises. Sustainability.” -Reviewer 1
Authors’ response:
These remarks helped to further improve Conclusions section. The answers to the defined research questions has been answered more clearly, i.e., the impact of legislative changes to the studied region’s business environment, new business models enabled in the context of this region, and the innovations enabling these business models. Especially the third one has been emphasized in the revised version. This made the contribution of the research to the literature more apparent. The provided reference article was helpful in aligning the manuscript better with the existing scientific literature. Through establishing the limitations more explicitly, it was possible to also communicate the contribution of this research as well as justify the proposed future studies.
